# Mass Spectrometry as a Quantitative Proteomic Analysis Tool for the Search for Temporal Lobe Epilepsy Biomarkers: A Systematic Review

**DOI:** 10.3390/ijms241311130

**Published:** 2023-07-05

**Authors:** Elena E. Timechko, Alexey M. Yakimov, Anastasia I. Paramonova, Anna A. Usoltseva, Nikita P. Utyashev, Nikita O. Ivin, Anna A. Utyasheva, Albina V. Yakunina, Vladimir A. Kalinin, Diana V. Dmitrenko

**Affiliations:** 1Department of Medical Genetics and Clinical Neurophysiology of Postgraduate Education, V.F. Voino-Yasenetsky Krasnoyarsk State Medical University, 660022 Krasnoyarsk, Russia; e.e.timechko@yandex.ru (E.E.T.); cilvana20010z@gmail.com (A.M.Y.);; 2Federal State Budgetary Institution “National Medical and Surgical Center Named after N.I. Pirogov”, 105203 Moscow, Russia; 3Department of Neurology and Neurobiology of Postgraduate Education, Samara State Medical University, 443079 Samara, Russia

**Keywords:** temporal lobe epilepsy, proteomic, biomarkers, mass-spectrometry, protein expression

## Abstract

Temporal lobe epilepsy (TLE) is the most common form of epilepsy in adults. Tissue reorganization at the site of the epileptogenic focus is accompanied by changes in the expression patterns of protein molecules. The study of mRNA and its corresponding proteins is crucial for understanding the pathogenesis of the disease. Protein expression profiles do not always directly correlate with the levels of their transcripts; therefore, it is protein profiling that is no less important for understanding the molecular mechanisms and biological processes of TLE. The study and annotation of proteins that are statistically significantly different in patients with TLE is an approach to search for biomarkers of this disease, various stages of its development, as well as a method for searching for specific targets for the development of a further therapeutic strategy. When writing a systematic review, the following aggregators of scientific journals were used: MDPI, PubMed, ScienceDirect, Springer, and Web of Science. Scientific articles were searched using the following keywords: “proteomic”, “mass-spectrometry”, “protein expression”, “temporal lobe epilepsy”, and “biomarkers”. Publications from 2003 to the present have been analyzed. Studies of brain tissues, experimental models of epilepsy, as well as biological fluids, were analyzed. For each of the groups, aberrantly expressed proteins found in various studies were isolated. Most of the studies omitted important characteristics of the studied patients, such as: duration of illness, type and response to therapy, gender, etc. Proteins that overlap across different tissue types and different studies have been highlighted: DPYSL, SYT1, STMN1, APOE, NME1, and others. The most common biological processes for them were the positive regulation of neurofibrillary tangle assembly, the regulation of amyloid fibril formation, lipoprotein catabolic process, the positive regulation of vesicle fusion, the positive regulation of oxidative stress-induced intrinsic apoptotic signaling pathway, removal of superoxide radicals, axon extension, and the regulation of actin filament depolymerization. MS-based proteomic profiling for a relevant study must accept a number of limitations, the most important of which is the need to compare different types of neurological and, in particular, epileptic disorders. Such a criterion could increase the specificity of the search work and, in the future, lead to the discovery of biomarkers for a particular disease.

## 1. Introduction

Temporal lobe epilepsy (TLE), characterized by the presence of recurrent seizures with onset localized in the amygdala–hippocampi–parahippocampal area [1], is the most common type of focal epilepsy in adults [2].

In most patients, the development of TLE was preceded by a head injury, a complex of febrile seizures, stroke, and episodes of ischemia or intracerebral infection [3,4]. The initiating event leads to the transformation of brain tissues and cells, as well as to a change in intracellular and extracellular molecular mechanisms. During this latent period of brain tissue transformation, as a rule, no clinical symptoms of the disease were observed [5]. Large-scale neuronal and synaptic reorganization of the affected tissue is associated with such processes as: neurodegeneration, gliosis, axonal sprouting, neuroinflammation, blood–brain barrier (BBB) dysfunction, oxidative stress, etc. [6].

It is also known that tissue reorganization in the epileptic focus is accompanied by changes in the expression patterns of protein molecules [7]. Thus, one of the classical methods for detecting astrogliosis was immunohistochemical staining of the affected tissue for the presence and prevalence of GFAP [8]. One of the approaches to the study of the molecular mechanisms of epileptogenesis is neuroproteomics, which is a large-scale study of the proteomic profile [9] of tissue and circulating proteins.

During the development of molecular biology, it became obvious that the processes occurring in the body at the genetic, cellular, and ontogenetic levels cannot be fully described based only on genomic studies.

Qualitative and quantitative detection of mRNA and corresponding proteins expressed in various tissues is critical to understanding human physiology and pathology. A significant amount of research has focused on measuring RNA levels. However, protein levels do not always directly correlate with the levels of their transcripts [10], which is why protein profiling is no less important for understanding molecular mechanisms. Proteomic analysis is aimed at the simultaneous study of many individual proteins, the total amount of which makes up a certain system, which characterizes the object under study as a whole.

Quantitative proteomics is a technique based on determination of the amount of protein in a target sample [11]. Proteomic analysis of samples derived from complex samples, based on mass spectrometry, provides a deeper understanding of molecular and cellular processes and functions. This approach can allow identification of the difference in the expression profiles of patients suffering from TLE and healthy people.

Mass spectrometry (MS) is one of the most important tools for proteomic analysis today. To date, MS underlies almost all proteomic experiments, as it provides key tools for protein analysis.

The study and annotation of proteins that are statistically significantly different in patients with TLE is an approach to search for biomarkers of this disease, various stages of its development, as well as a method for searching for specific targets for the development of a further therapeutic strategy.

The purpose of this review is to analyze the results obtained and described in existing publications, to indicate the limitations of these studies, and to outline strategies for further search for biomarkers of epileptogenesis and/or temporal lobe epilepsy.

## 2. Methods and Research Design

When writing a systematic review, the following aggregators of scientific journals were used: MDPI, PubMed, ScienceDirect, Springer, and Web of Science. Scientific articles were searched using the following keywords: proteomic, mass-spectrometry, protein expression, temporal lobe epilepsy, and biomarkers.

Scientific articles from 2003 to 2023 were analyzed, including original research and review articles. Publications were included in the systematic review if they were published in English and complied with the inclusion criteria defined below, and duplicates were excluded from the review. The process of searching for publications for a review is shown in Figure 1.

Bioinformatic analysis was carried out using such platforms as: UniProt (https://www.uniprot.org, accessed on 20 April 2023), ShinyGO (http://bioinformatics.sdstate.edu/go/, accessed on 25 April 2023), StringDB (https://string-db.org, accessed on 15 April 2023), and DisGenNet (https://www.disgenet.org, accessed on 30 April 2023).

### 2.1. Inclusion Criteria

The different laboratory methodologies and samples utilized across the studies are summarized in Figure 2.

#### Study Material

Studies of proteomic investigation of TLE were included in the review. Studies of patients with TLE were limited by the patient’s age. No limitations were placed on ethnicity, country of origin, family history of recurrent miscarriage, or any other patient demographics.

Material for patients’ investigation had no limitation, and all samples were available: brain tissues, cerebrospinal fluid, plasma, and blood.

Laboratory methods for identifying protein expression profile included in the review were:-Mass-spectrometry (MALDI; MALDI/TOF: MALDI/TOF-TOF; ESI; etc.) with and without labels.-Two-dimensional gel-electrophoresis + mass-spectrometry.-Western blot + mass-spectrometry.

### 2.2. Exclusion Criteria

Studies of patients with TLE were limited by patient’s age. Studies on children were excluded. Studies of different forms of epilepsy were excluded from the review.

The absence of mass-spectrometry in the methods was the main exclusion criterion.

## 3. Results and Discussion

One of the main and most traditional methods of proteomic analysis is two-dimensional gel electrophoresis in polyacrylamide gel [12]. The essence of the method is based on the ability to separate proteins based on their physical and chemical properties, such as molecular weight and isoelectric point. The first measurement during two-dimensional gel electrophoresis is the separation of proteins on immobilized strips with a pH-gradient according to the isoelectric point (pI). During the second measurement, using electrophoresis in SDS-polyacrylamide gel, the denatured proteins are separated according to their molecular weight [13]. Initially, the disadvantage of the method was the inability to identify protein spots of interest, i.e., those that showed differential expression. Moreover, difficulties were also observed for poorly soluble, hydrophobic proteins.

Therefore, as computer hardware and software improved, it became possible to identify proteins using mass spectrometry.

In many studies, proteomic profiling was carried out based on a combination of two methods, two-dimensional gel electrophoresis and mass spectrometry, where the MS-method was used to identify protein spots with established differential expression [14].

More recently, mass spectrometry has become increasingly used as a stand-alone tool for proteomic profiling, separated from two-dimensional electrophoresis.

As a rule, proteomic analysis, based on mass spectrometry, is divided into two main approaches: “top-down” proteomics, in which intact proteins are studied, and “bottom-up” proteomics, in which the protein is first degraded to peptides, and the latter are already subjected to measurement [15]. The specific structure of the fragment ion of each peptide ion, together with its *m*/*z* value, makes it possible to accurately identify the peptides present in the sample. Then, the identified peptide sequences can be compared with proteins, and the signal intensity of peptides or fragment ions can be used to assess the relative changes in their content in samples [16].

The gold standard of MS analysis is mass spectrometry using isotope labels: TMT-labels, iCAT and iTRAQ [15]. However, today, label-free MS has become more popular due to simpler and cheaper sample preparation [17].

The search for biomarkers of the disease, the epileptogenic process, and/or its stage is the main goal of proteomic profiling. However, this task faces a number of difficulties. The results obtained during the MS analysis describe «static data», while aberrant protein expression is often temporary, which is what needs to be taken into account when designing and setting up the experiment [18]. Among other things, differential expression may not always be a direct reflection of the pathogenic process, for example: there are a number of correlations between the protein profile and a number of parameters unrelated to the disease. For example, there are correlations between a set of hippocampal-specific proteins and gender [7].

In addition, it is important to take into account that the stage of the disease, as well as the length of the pathology, can have a serious impact on protein expression. In experimental models, a dependence was found between the expression of a number of proteins and the stage of the course of the disease [19,20], in particular: different stages of pathogenesis can demonstrate not only quantitatively, but also qualitatively, different expression patterns. Additionally, in our earlier study [21], a significant decrease in *TNFa* expression in the plasma of patients with TLE with a disease experience of more than 10 years was found, compared with patients with a shorter disease experience. Additionally, most studies do not indicate the age of patients and the duration of the disease. However, these parameters may be significant and affect the proteomic profile. For example, in a study by Liu (2020) [22], it was shown that most disease-specific proteins are found in older patients (less than 50 years): 19% of identified proteins were observed only in a younger cohort (less than 35 years), and 28% were observed only in older samples. Additionally, Zhang (2020) [23] took into account the effect of age on expression, and, according to his data, only 2.18% of protein expression varied in age-differentiated cohorts, from which it was concluded that there was no statistically significant correlation between age and expression profile. Some studies [24] took into account the age of patients and the duration of the disease, but its effect on expression was not evaluated. The difference in protein expression was seen in the fold change and the ratio of molecular weight to isoelectric point (pI/mW).

### 3.1. Tissues Profiling

The hippocampus and parahippocampal area are the primary substrates of the epileptogenic process [25], so the study of these tissues is the most common and significant. As a rule, the object of proteomic analysis is a hippocampal biopsy of patients undergoing surgical treatment due to the development of refractoriness to antiepileptic drugs.

The use of biopsies also leads to difficulties in finding a biomaterial for the control cohort. Surgically resected “normal” hippocampal tissue of healthy people is not available for research [22], which leads to the need to study autopsy material or samples of brain tissue affected by another disease. However, the protein extraction process for proteomic analysis differs between surgical and post-mortem brain tissue. Therefore, the use of the same protocol for studies can lead to irrelevant results. Among other things, autopsy tissues can be subjected to post-mortem transformation processes, and many proteins, such as hemoglobins, as well as cytoskeletal component proteins, can quickly degrade, and mutant forms of the protein can also appear [26]. All of the above can significantly affect the results.

Additionally, it is important to understand that hippocampal tissue samples for research are obtained from patients undergoing surgical treatment due to the development of refractoriness to antiepileptic drugs. There is evidence of the presence of alternative RNA expression profiles between groups of drug-sensitive and drug-resistant patients [27], which will also be reflected in the protein profile. Additionally, brain tissue with the presence of structural changes, such as hippocampal sclerosis, is most often examined, the presence of which is a typical, but not mandatory, phenomenon in TLE [28]. These factors are important to take into account in the search for TLE biomarkers, since protein profiles may be characteristic of these features and conditions of TLE. Taking into account all the characteristics of the studied patient is an important and necessary stage in the design of the experiment.

Additionally, it is important to record the presence and type of pharmacotherapy because AEDs can affect the expression of certain proteins [29,30]. In particular, one of the classic AEDs, carbomazepine, affects the expression of UDP-glucuronosyltransferase [31]. Such limitations should be taken into account and also indicated when selecting potential candidates for biomarkers of epilepsy or epileptogenesis. In some cases, aberrant expression may not be an indicator of a pathogenic process, but only the result of therapy.

In the earlier work on proteomic profiling of tissues, there was still a trend towards the combined use of two-dimensional electrophoresis and mass spectrometry. Additionally, research was often targeted, and it was aimed at studying a specific group of proteins involved in a particular biological process associated with epilepsy.

Thus, two-dimensional gel electrophoresis, coupled with MALDI-TOF analysis of brain cortex proteins in patients with TLE, was performed by Eun (2004) [32]. Images of the non-epeleptogenic temporal lobe of patients undergoing surgical treatment for glioblastoma multiforme served as controls. The study revealed a clear relationship between the levels of the antioxidant enzyme Mn-SOD (MnSOD) and the pathogenesis of TLE. The level of Mn-SOD was consistently reduced by 45 ± 9% in the cerebral cortex of patients with epilepsy, compared with that of patients without epilepsy. Additionally, four proteins were found to be consistently overexpressed in all epileptic tissue samples from a patient’s brain compared to non-epileptic samples: UMP-CMP kinase (CMPK2), proteasome activator complex subunit 2 (PSME2), glyceraldehyde 3-phosphate dehydrogenase (GAPDH), apoptosis regulator BAX, and cytoplasmic isoform beta (BAX). The other four proteins were found to be reverse-expressed: GPDH, eukaryotic peptide chain release factor subunit 1 (eRF 1), glycerol-3-phosphate dehydrogenase [NAD+], cytoplasmic, carbonic anhydrase VII, and DNA primase large subunit. The identified proteins have been proposed as early disease biomarkers. It is important to point out that, in other studies of astrocytic tumors, the level of Mn-SOD also significantly decreases [33,34].

Two-dimensional electrophoresis, combined with MS (MALDI-TOF), was also presented by Yang (2006) [35]. In this study, 77 proteins were identified in hippocampal samples from patients with mTLE and controls (autopsy). Proteins were quantified, and 18 proteins with mTLE differential expression were observed, and among them, several classes were distinguished: antioxidant proteins (peroxiredoxins 3 and 6), chaperones (T-complex protein 1 subunit alpha, stress-induced phosphoprotein1), signal proteins (mitogen-activated protein kinase kinase 1 (MAP2K1)), synaptosomal proteins (synaptotagmin I (SYT1), α-synuclein (SNCA)), NAD-dependent deacetylase sirtuin-2 (SIRT2), protein 7 of the regulatory subunit of the 26S protease (PSMD7), and neuron-specific septin 3 (SEPT3). The most significant finding was a significant decrease in the expression of cytoskeletal proteins: tubulin α-1 (TUBA1A), β-tubulin (TUBB), profilin II (PFN2), and neuronal tropomodulin (TMOD2). Patients with drug-resistant TLE and hippocampal sclerosis were studied, and the type of therapy was also provided, but the effect of therapy on protein expression was not taken into account.

Persike (2012) [36] performed a proteomic analysis of hippocampal samples from patients with TLE compared with autopsy controls based on two-dimensional gel electrophoresis, coupled with MALDI-TOF-MS. Nine differentially expressed proteins were found, including: proton ATPase catalytic subunit A (ATP6V1A) and dihydrolipoyllysine-residue acethyltransferase, a component of the pyruvate dehydrogenase complex (DLAT), as well as HSP 70, dihydropyrimidinase-related protein 2 (DPYSL2), isoform 1 of serum albumin (ALB), isoform 1 of myelin basic protein (MBP1), overexpressed, glutathione S-transferase P (GSTP1), and DJ-1 (PARK7), found only in the TLE hippocampus, and isoform 3 of spectrin alpha chain (fodrin) (SPTAN1), which was hypoexpressed. According to the results of their GeneOntology analysis, it was revealed that specific proteins are involved in the mechanisms of compensation from oxidative stress and excitotoxicity, cytoskeletal remodeling, and Ca^2+^ homeostasis.

Keren-Aviram (2018) [37] analyzed human brain biopsies taken from the neocortex of six patients with refractory epilepsy (the exact diagnosis was not specified). A distinctive feature of this study is the refusal to use autopsy material as a control: a comparison was made of protein expression in the same patient—a more epileptogenic area was compared with a less epileptogenic area of the brain. Proteins were separated from three subcellular fractions using two-dimensional differential gel electrophoresis (2D-DIGE), and 31 protein spots were identified with significant aberrant expression for the experimental cohort. Thirty-one of these spots were identified by ESI-MS-MS as coming from 18 gene products, some of which existed in multiple isoforms. In epileptogenic samples, eight gene products were overexpressed (SNCA, STMN1, UGP2, DSP, CA1, PRDX2, SYN2, and DPYSL2), and ten were downexpressed (GFAP, HNRNPK, CPNE6, CRYAB, GNAO1, PHYHIP, HNRPDL, ALDH2, GAPDH, and LASP1). Many of the identified proteins belonged to the cytoskeleton, which, according to the findings of Keren-Aviram, indicates structural or migratory changes in the affected cells. Dominant in this group of proteins with statistically significant downregulation in high peak regions in all fractions for all patients was GFAP (~50 kDa): a commonly used astrocytic marker.

In Mériaux (2014) [7], there has already been a transition to the use of MS as an independent method. Thus, MALDI-MS with LFQ was performed to analyze the dentate gyrus of the hippocampus of patients with TLE. For the first time, the effect of gender on expression was taken into account, so specific proteins found in patients with temporal lobe epilepsy account for 27.1% in men and 19.6% in women. The comparison of the control and TLE-specific proteins by sex showed that samples obtained from men with TLE contained 29.9% specific proteins, while samples obtained from women with TLE contained less specific proteins (21.5%). An amount of 357 proteins were found in women with TLE, and 604 were found in men with TLE. Based on the score in the LFQ, 40 proteins, characteristic of the gender-mixed group of patients with TLE, were selected, including: LGI1, TUSC2, SRC8, CTCF, and others. Most of the identified proteins are involved in neurite outgrowth, neuronal differentiation, cytoskeletal network organization, cell signaling, and tumor repression (DMBT1, TUSC2, MGEA5; GBAS, CNDP2, and LIGI1). Tellingly, most tumor suppressors (TS) have only been identified in male patients. In men with TLE, among the identified TSs, the LGI1 protein was found, which is usually expressed in the developing and adult CNS, especially in the neocortex, granular cells of the dentate gyrus, and the CA3-CA1 region of the hippocampal pyramidal cell layer [38]. It should be taken into account that TS is associated with tumor diseases and a number of neurological disorders [39]. Secretogranin-1 (CHGB1), secretogranin-2 (CHGB2), galanin (GAL), appetite-regulating hormone isoform 2 (GHRL), angiotensin (AGT), and neuroendocrine convertase 2 (PCSK2) have also been found at high levels in male TLE, while proenkephalin B (PENK) and growth hormone 2 (GH2) were specifically found in women with TE. Regarding their receptors, the male TLE specifically contained the NPY type 2 receptor (NPY2R), vasopressin (ADH), as well as the mu-opioid cut receptor (OPRM1). Female TLEs highly express VIP receptors and leptin receptors, from which the authors concluded that endocrine hormones and their receptors are actively involved in the development of temporal lobe epilepsy.

Liu (2020) [22] compared the proteomic profiles of granular cells (GCs) of the hippocampus: basal and dispersed GCs. The influence of age on protein profiles was also taken into account: a cohort of younger (under 35 years old) and older (over 50 years old) subjects. It is believed that the dispersion of granular cells is a characteristic pathological feature in TLE [40,41]. Therefore, the expression profiles of proteins involved in the reactivation of neurodevelopmental migration pathways will differ in pathology and normal physiological state. A proteomic analysis, based on MALDI-MS-LFQ of hippocampal biopsy specimens from patients with mTLE + HS undergoing surgical treatment, was performed. Granular cell dispersion was confirmed by immunohistochemical staining. In a proteomic analysis of eight patients with type 1 HS and granular cell dispersion, 1808 proteins were identified. An amount of 54% of the studied proteins were found in both types of samples, 29% of these proteins were found only in basal samples, and 17% were found only in dispersed samples. An amount of 19% of the studied proteins were found only in the cohort with younger patients, 28% were found only in the cohort with older patients, and all other detected proteins did not correlate with the age of the patients. In the general cohort of patients with mTLE, neuronal markers were found, such as: MAP2, calbindin (CALB1), and calretinin (CALB2). For MAP2, CALB1, and CALB2, there was a correlation with the age of the patients—their level was significantly higher in the cohort with younger patients. Astroglial (GFAP, vimentin (VIM), S100-B) and oligodendroglial markers (myelin basic protein (MBP)) were also detected in all samples. GFAP expression was comparable between basal and dispersed samples and was slightly higher in younger than in older samples. Interestingly, the expression of vimentin, expressed in immature astrocytes, was observed predominantly in basal and younger samples. A higher content of cell adhesion molecules 2 (CADM2), a synapse-associated protein found in the subventricular neurogenic niche, as well as a number of cell cycle markers (MCM2, PCNA, and cyclin B3), were found in higher amounts in basal GCs than in dispersed GCs. The best clusters of functional annotations included GTP binding and activity, intercellular adhesion, and low GTPase-mediated signaling. Dispersed GCs highly express markers of mature neurons, MAP2, CALB1, and CALB2, as well as a number of Rho GTPases and proteins associated with cell migration, the cytoskeleton, and synapse remodeling.

API-TMT-tagged MS was performed in a study by Zhang (2020) [23]. Biopsy specimens of patients with mTLe + HS undergoing surgical treatment were analyzed, and the mean age of the patient was 34.0 ± 10.5; autopsy specimens served as controls. In both cohorts, control and experimental, 3023 proteins were found. Comparison of the brain tissues of TLE patients and controls revealed 211 proteins with statistically significant altered expression. Among the identified proteins, an increase in expression was found for 141 proteins, as well as a decrease in expression for 70 proteins. The expression of proteins involved in synaptic vesicular transport, such as ATP1A2, SLC25A4, EAAT1 (SLC1A3), VGLUT1 (SLC17A7), RAB3A, SYT1, STX1A, STX1B, and SYP, was significantly reduced in TLE patients compared to controls. The S100A10, S100A6, ANXA1, ANXA2, ANXA5, and AKR1C3 prostaglandin synthesis and regulation pathways were markedly elevated in the TLE hippocampus compared to levels found in control tissue. Activation of several members of the complement cascade (C4A, C4B, CFH, CFB) has also been found in the TLE hippocampus. The obtained data were validated by Western blotting and immunohistochemical staining.

The work by Xiao (2021) [24] was devoted to proteomic profiling of the dentate gyrus (DG), made by API-Q-MS. An isobaric label for relative and absolute quantitation (iTRAQ), based on a quantitative proteomic method, was used to analyze hippocampal DG obtained from TLE-HS patients compared to autopsy controls. The mean age of patients with mVE + HS = 25.2 ± 9.2, and the mean disease duration = 8.8 ± 5.4. An amount of 5583 proteins were identified, of which 82 proteins were activated, and 90 proteins were downregulated. Bioinformatic analysis showed that differentially expressed proteins were enriched in the “synaptic vesicle”, “cell–cell adhesion”, “regulation of synaptic plasticity”, “ATP binding”, and “oxidative phosphorylation”. Analysis of the network of protein–protein interactions revealed a key module of 10 proteins associated with “oxidative phosphorylation”. The hypoexpressed proteins included such proteins as: LRRFIP2, LACTB, SON, PAK2, etc., as well as the overexpressed ones—AIDA, BAX, ATP5ME, ATXN2, etc. Bioinformatics analysis of KEGG pathways revealed that many of the detected proteins are involved in oxidative phosphorylation and metabolic pathways. Among the proteins involved in these processes are: NDUFA5, UQCR10, NDUFS5, ND4, and ATP5I.

It is important to note that, in our work, we considered only those proteins that were listed in the original publication and/or in the Supplementary available for free download. In the course of studying publications on the topic, we conducted our own bioinformatics analysis of all proteins studied in tissues [7,22,23,24,32,35,36,37] using ShinyGo 0.77 (bioinformatics.sdstate.edu/go/, accessed on) (Figure 3).

The involvement of the entire cohort of proteins studied in various publications in the cycle of synaptic vesicles, signaling, and transmission modulation can be distinguished from the characteristic features.

If we analyze the intersections of proteins found in various studies, then only some of them will be common (Table 1).

As can be seen from the figure, matching proteins turned out to be: AIDA, DPYSL2, GFAP, SYT1, SNCA, GAPDH, and BAX. GO analysis demonstrated their involvement in the regulation of apoptosis. For a complete list of protein dysregulations found, see Appendix A.

For many of the cases studied, experimental tissue samples were affected by sclerotic changes. Additionally, hippocampal sclerosis was often accompanied by refractoriness to treatment with antiepileptic drugs. Therefore, it becomes difficult to determine which of the detected proteins reflect histological changes in the tissue, and which are involved in the genesis of epileptic seizures. These facts must be kept in mind as limitations of the research.

### 3.2. Experimental Models

The study of protein expression is complicated by the inaccessibility of brain tissues: obtaining a sample of hippocampal tissue is available only in cases of surgical intervention or after death. Therefore, in proteomic studies, experimental animal models are often used as an analogue of the conditions of the human body. However, it is important to keep in mind that experimental models are quite limited, which can lead to the appearance of artifacts. In addition, it must be taken into account that homologous proteins of different organisms often do not have the proper degree of similarity. For example, mouse APOE, a protein studied for various neurodegenerative diseases [42], according to UniPROT (www.uniprot.org, accessed on 20 April 2023), does not even have 90% similarity with human APOE.

Nevertheless, experimental models, to one degree or another, shed light on the molecular mechanisms of the pathogenesis of the disease. From models, it is easy to screen hundreds of candidate biomarker molecules in a short time. At present, post-status models, such as pilocarpine or kainate or pentylenetetrazole models, are the most widely used models for studying epileptogenic processes and drug targets that can prevent or modify epilepsy [43].

The advantage of using models is the ability to track changes in protein expression profiles at different stages of disease pathogenesis. Thus, the search for invariant biomarkers of epilepsy can be performed by studying protein expression at all three stages of TLE development: acute, latent, and chronic phases.

The search for TLE-specific aberrantly expressed proteins, based on the MS method, was carried out in a number of studies using brain tissues of animal experimental models.

One of the first proteomic analyses of the tissues of the hippocampus of an experimental animal can be considered in the study by Liu (2008) [44], in which, during MALDI MS and MS/MS analysis, 57 differentially expressed proteins were identified that were either activated or suppressed at two time points: 12 h (acute period) and 72 h (latent period) compared to the control. In the acute phase, among proteins with reduced expression, 24 proteins were found, including: Acta1, Vim, Tubb2b, Hspd1, etc., as well as 19 overexpressed proteins—Hsp90ab1, Nef3, Syn2, Vdac2, etc. During the latent period, the number of aberrant proteins decreased among proteins with reduced expression, and 12 were found: Dlat, Eno1 Hspd1, Hba2, and 19 were overrepresented—Homer2, Ddt, Syn2, Vdac2, and others. Invariants for both periods were: Hspd1, Snap25, Glo1, Arhgdia, Nme1, Eno1, Hba2, Hbb (hypoexpression), as well as Homer2, Ddt, Syn2, Ina, Msi1, and Cox6b1 (hyperexpression).

Araujo (2014) [45] performed a proteomic analysis by ESI-Q-TOF-MS of the dentate gyrus (DG) in rats with pilocarpine-induced epilepsy. The study was carried out using two-dimensional gel electrophoresis and tandem MS. An amount of 33 differentially expressed proteins were identified, most of which were suppressed in the tissues of experimental animals: Ckb, Suclg1, Tpi1, Oxct1, and others. The tissue for analysis was taken 30 days after the onset of the first spontaneous attack. Thus, the expression of acute phase proteins was not assessed.

Additionally, in a study by Bitsika et al. (2016) [19], the ESI-LC-MS/MS method, in combination with LFQ, revealed 22, 53, and 175 statistically significant differentially expressed proteins in the tissues of mice brain exposed to kainate in the 1st, 3rd, and 30th day after the induced status, in comparison with control samples of the brain of mice that were injected with NaCl. The effectiveness of the development of signs of the disease was established by observing the behavioral characteristics of the animal, as well as by measuring the level of GFAP by immunohistochemistry, which is a marker of astrogliosis [46]. Most of the aberrantly expressed proteins of the acute phase of epilepsy were suppressed in the experimental cohort compared with the control: Neurabin-2 (Ppp1r9b), Neurochondrin (Ncdn), and Homer1. Meanwhile, some were found only in the control samples: Ataxin-10 (Atxn10), Shank3, and WD repeat-containing protein 47 (Wdr47). In the latent phase, a different distribution was found: 21 overexpressed proteins, including moesin (Msn), clusterin (Clu), vimentin (Vim), GFAP, and 32 underexpressed proteins. These included neurabin-2 (Ppp1r9b), brain-specific angiogenesis inhibitor 1-associated protein 2 (Adgrb2), Src substrate cortactin (Cttn), and sodium- and chloride-dependent GABA transporter 1 (Slc6a11), and some were found only in controls. These include brain-specific serine/threonine-protein kinase 1 (Brsk1), A-kinase anchor protein 5 (Akap1), and Shank3. In the chronic stage of TLE, most of the detected proteins were either overexpressed or found only in experimental samples: clusterin (Clu), GFAP, vimentin (Vim), gelsolin (Gsn), and apolipoprotein E (Apoe). The following turned out to be invariant for each stage: neurabin-2 (Ppp1r9b), whose expression was suppressed by 0.3–0.4 times at each measurement stage, serotransferrin (Trf), overexpressed 3–3.7 times, Src substrate cortactin (Cttn), hypoexpressed 0.47–0.13-fold (on the 1st and 3rd day, as well as the 30th day, respectively), and microtubule-associated protein 2 (Map2), suppressed by 0.43–0.53 times.

A study by Walker (2016) [47], in the same year, on rats using ESI-MS/MS based on LFQ already took into account not only temporal expression patterns, but also different regions of the brain temporal lobe: the hippocampus and the parahippocampal area (PHA). The authors focused on the proteins involved in the inflammatory process and the immune response. Therefore, the total number of aberrantly expressed proteins found is not reported, which significantly complicates the analysis and does not allow for a full-scale study. The presence of status epilepticus and epilepsy was confirmed by EEG monitoring. Two days after the electroshock-induced status, invariant dysregulation was observed: heat shock proteins, Msn, Actn1 (overexpressed for both tissues), and tollip, as well as Cdc42-202 (hyper and hypo for the hippocampus and PHA, respectively), in the hippocampus and the parahippocampal region. In general, a greater number of aberrantly expressed proteins were found for PHA. In the latent phase (10 days after the status) for both regions, aberrant expression of such proteins as heat shock proteins, P2R proteins, Itg, Serpinh1, Actn1 (hyperexpressed for both tissues), and proteins of the Dnm family, such as Usp7 (hyper and hypo for hypocampus and PHA, respectively), were observed. In the chronic phase, proteins invariant for both regions were not found.

Keck (2017) [20] analyzed the hippocampus and parahippocampal area (PHA) by ESI-MS/MS. Tissue type and temporal expression patterns were taken into account. Thus, the tissues of the temporal lobe were studied 2, 10, and 56 days after kindling-induced status epilepticus. Differential expression analysis of hippocampal proteins showed that 121 proteins were differentially expressed at two days, 276 proteins were differentially expressed at ten days, and 14 proteins were differentially expressed at eight weeks after SE. It turned out that, in PHA, 218, 419, and 223 proteins are expressed differently two days, ten days, and eight weeks after SE, respectively. Differentially expressed proteins at all three time points showed an overlap of four hippocampal and twenty-three PHA proteins. The amount of proteins invariant for the acute phase and the latent period for PHA was 129, and, for the hippocampus, it was43. The number of proteins invariant for the latent phase and chronic period for PHA was 41, and, for the hippocampus, it was 3. The common proteins for PHA and the hippocampus were twenty-seven proteins in the acute phase, eighty-seven proteins in the latent period, and eight proteins in the chronic phase. PCA analysis revealed the top 10 proteins for each of the brain regions at various time intervals. In the acute phase, the greatest aberration (fold change ≥1.5 (upregulated in SE samples) or fold change ≤0.67 (down-regulated in SE samples) was found for: Cd151 molecule, mannose-P-dochilol utilization defect 1 protein (Mpdu1), and clusterin (Clu). In the chronic phase following epilepsy manifestation, four proteins were overexpressed: H+ transporting mitochondrial complex Fo, proteasome type 3 beta subunit (Psmb3) purine nucleoside phosphorylase (Pnp), ATP synthase, and glutathione S-transferase alpha (Gsta2). The most prominent molecular changes in the respective tissue types were detected during the latent phase.

A study by Xu (2020) [48] identified 99 aberrantly expressed proteins in the hippocampus of rats with pentylene-tetrazole-induced epilepsy, and quantitative proteomic analysis was performed using ESI-Q-LC/MS iTRAQ-labeled quantification. An amount of 93 proteins were overrepresented in the experimental hippocampus, including Rttn, Amfr, Slc30a1, S100a4, and Syvn1, and six were underrepresented, including Iqca1, Tmprss13, Prss1, Banf1, and Ptp.

A two-dimensional electrophoresis + MALDI-TOF/TOF-associated study of the kainate and pilocarpine rat hippocampal proteome was performed by Sadeghi (2021) [49]. The hippocampal proteomic profile of model rats was compared with a control proteomic profile (healthy hippocampus), taking into account laterality. During the study, 144 proteins were detected. Among them were 95 proteins that were found that were common to the experimental and control groups. These 95 proteins were further digested with trypsin and MALDI-TOF-TOF/MS analysis. Among the identified proteins were glutamate NMDA receptor subunit epsilon-1 (Nmde1), Adprc, Lpar3, glutamate AMPA receptor R1, calreticulin, and S100 calcium binding protein A3, which are overexpressed. Also found were snap, transmembrane protein 165, and Rcn2, which are are hypoexpressed. The protein expression profiles of both hippocampuses were studied to determine if seizures would change protein expression asymmetries.

In Qian (2022) [50], ESI-MS/MS analysis with an isobaric label was performed on hippocampal tissues only two weeks after epilepsy induced by lithium-pidocarpine, which was characterized as the chronic stage of epilepsy. In the course of the work, twenty-seven differentially expressed proteins were found out of four thousand one hundred and seventy-three detected, including eighteen proteins with increased expression and nine proteins with reduced expression (*p* < 0.05 fold change ≥1.5—upregulated in SE samples or fold change ≤0.67—down-regulated in SE samples). Among the overexpressed proteins were Gfap, Camk, and Krt family proteins, as well as Nudc, Cadm1, Vim, Cd9, Abcf1, Fnta, Dnajc3, Vps53, Ogfr, Eipr1, and Cibar2; among the hypoexpressed proteins were Nrgn, H4c2, H2B1, H2ax, H2ac18, Prrt3, and Crym.

The overlaps of the detected proteins are presented in Table 2.

The analysis of biological and molecular functions of studied proteins [19,20,44,47,48,49,50] were based on GeneOntology, made using ShinyGo 0.77 (Figure 4), and it revealed a number of the following functions (at FDR = 0.05): synaptic signaling, neurogenesis, synaptic organization, organization of the cytoskeleton, and the regulation of transport functions.

### 3.3. Circulating Proteins

Proteomic analysis of biological fluids of patients with TLE is based on the study of blood plasma, serum, or cerebrospinal fluid (CSF). To date, there are few studies focusing on the analysis of circulating proteins, as compared to tissue proteomic analysis, and CSF testing can reveal the molecular and biochemical changes that occur in the brain in the presence of a neurodegenerative disease. However, obtaining CSF is an invasive procedure—involving a lumbar puncture—requiring a doctor’s prescription. Blood plasma is a more convenient and simple sample to study because the collection procedure is less invasive, less time-consuming, and more routine [51].

Blood plasma is also a more suitable sample for population-based studies [52]. Plasma offers some advantages, such as reflecting markers of tissue leakage, but it does not detect biomarkers that are not secreted by the CNS. Blood plasma may represent the most extensive version of the human proteome, since, in addition to classical plasma proteins, it also contains tissue proteins—arkers of “tissue leakage” [53]. Tissue leakage markers are proteins that, under normal physiological conditions, function inside the cells of various tissues, but they are released into the blood plasma when cells are damaged and degenerated. Among the markers of tissue leakage are known diagnostic biomarkers of myocardial infarction: cardiac troponins, creatine kinase, or myoglobin. However, plasma can be problematic due to the wide range of protein concentrations.

Xiao (2009) [54] analyzed the CSF of patients with temporal lobe epilepsy (TLE) and the control group. Protein expression was studied using two-dimensional gel electrophoresis, followed by liquid chromatography, as well as ESI-tandem mass spectrometry. Gel electrophoresis results revealed five protein spots with statistically significant altered expression, and, among them were cathepsin D (CTSD), apolipoprotein J (APOJ), Fam3c, and superoxide dismutase 1 (SOD1)—with reduced expression in patients with TLE and vitamin D-binding protein (DBP)—with increased expression in patients with TLE. Six protein spots unique to TLE patients were also identified: tetranectin (TN), talin-2 (TLN2), apolipoprotein E (APOE), immunoglobulin lambda light chain (IGL), immunoglobulin kappa variable light chain 1-5 (IGKV1 -5), and C-endopeptidase 1 procollagen enhancer (PCOLCE). A validating Western blot was performed for DBP, SOD1, and talin-2.

Banote (2021) [55] identified and mass-spectrometrically analyzed candidates for biomarkers of epileptogenesis/epilepsy in a group of adult patients with primary seizures made by API-Q-orbitrap-MS. Plasma was collected from 55 patients with first seizures, and follow-up was sufficient to detect epilepsy for follow-up testing. The subsequent two-year follow-up of patients led to the diagnosis of epilepsy in 63.6% of patients from the study group, and, in 36.4% of patients, only one seizure was registered. For the subsequent analysis of expression, blood plasma of these patients was extracted and labeled with tandem mass labels, and then it was analyzed using mass spectrometry. A total of 1075 proteins were identified, of which 41 proteins showed significant altered expression (expression that differed by ≥20% and *p* value < 0.05 was considered altered). A total of 41 proteins were significantly altered, of which 21 proteins were activated: MAPRE2, MBL2, PTX3, and LSP1, and 20 proteins were downregulated, and these included ENDOD1, NME1, and PDLIM1. The analysis of biological processes according to Gene Ontology revealed that the majority of proteins with an altered expression profile are involved in such processes as cellular protein metabolism and gene expression regulation. They also include proteins involved in the innate immune response, reorganization of the extracellular matrix, and the degranulation of neutrophils.

Both studies described above did not take into account the age of patients, the age of onset, and the duration of the disease. In addition, no studies have been conducted on the correlation of the use of AEDs, although Xiao indicated the type of therapy.

Hamrah (2020) [56] was the first to attempt to discriminate patients with TLE from patients with another neurological disease: psychogenic non-epileptic seizures (PNES), and this was performed not only in relation to healthy volunteers. This study was conducted in children, but we felt it was important to include it in the review because it is the only study that attempted to discriminate epilepsy from another disorder, not just healthy controls. The study included four patients with TLE, as well as four patients with PNES as a comparison group. Venous blood samples were taken no more than an hour after the seizure. Proteins from the blood were extracted and purified according to a standard protocol, and then they were subjected to separation using a two-dimensional eleutrophoresis gel. Differentially expressed protein spots were further analyzed using time-of-flight laser ionization using a matrix (MALDI/TOF) and electrospray ionization (MS) quad mass spectrometry. In total, 361 proteins were identified, among which 87 proteins were significantly upregulated (malate dehydrogenase 2 (MDH2)), and 110 proteins were downregulated in the TLE patient group (alpha 1 acid glycoprotein (ORM1), ceruloplasmin (CP), and S100-β) compared with a group of patients with PNES.

We also decided to analyze a publication on the study of the protein profile of patients with another form of epilepsy. This study is included in the review to compare and to search for intersections of protein expression in various forms of epilepsy. A study by Sun (2020) [57] used ESI-MS-TMT-based proteomics and bioinformatics analysis to identify protein expression profiles in the blood plasma of children with Rolandic epilepsy. A group of patients with Rolandic epilepsy was compared with a group of patients with migraine used as a control. Qualitative and quantitative analysis of proteins was carried out using liquid chromatography and TMT-mass spectrometry. The total number of identified proteins was 752, among which 217 were found to have altered expression. Forty-six identified proteins were overexpressed, and 111 were underexpressed in patients with Rolandic epilepsy compared with a group of patients with migraine. According to Sun (2020) [57], the development of epilepsy can be caused by the activation of such pathways and processes as: the activation of the acute phase or innate immune response, complement and fibrinogen systems, as well as suppression of glycolysis, lipoprotein metabolism, and antioxidant activity. The first GO terms were proteins associated with immune or inflammatory response, demonstrating yet again the result that, during the initiation and progression of epilepsy, there was a significant enrichment of immune or inflammatory response pathways [58,59]. Glycolytic enzymes may contribute to the development of the epileptogenic process. For example, downregulation of GAPDH kinase at γ-aminobutyric acid type A receptors (GABAARs) can reduce both endogenous phosphorylation and GABAAR function.

The matching proteins for each of the studies are shown in Table 3.

As can be seen, several proteins are aberrantly expressed in both TE and other types of epilepsy: MBL2, VWF, APOC1, SOD1, CTSD, ORM1, and PCOLCE. For SOD1, ORM1, and PCOLCE, even the same patterns are observed.

### 3.4. Discussions

Diagnostic biomarkers imply some staging invariance. It is assumed that the biomarker of the disease should be characteristic of a particular pathological condition and its differentiated expression should be observed from time to time.

A number of studies have found protein dysregulations characteristic of the proteome of TLE patients. A general analysis of all proteins, tissue (both models and patients) and circulating, gave the following pattern of overlaps of proteins found in studies [19,20,22,23,32,35,36,37,44,45,47,50,54,55] (see Table 4).

We considered the most significant proteins whose aberrant expression was observed in at least three different studies on proteomic profiling of TLE. In the case of proteins obtained in the study of the model, their human homologues were studied. A list of all matching proteins can be found in Appendix A.

Protein dysregulations that recur from study to study may be potential candidates for diagnostic biomarkers of epilepsy. Therefore, it was decided to analyze the studied proteins in order to evaluate their contribution to the epileptogenic process, as well as their specificity.

Analysis of all intersecting proteins in the STRING database (https://string-db.org, accessed on 15 April 2023) gave the following picture of the network: Figure 5.

Clustering these proteins into three cohorts revealed the following distribution by biological processes:

Green—positive regulation of neurofibrillary tangle assembly, regulation of amyloid fibril formation, Lipoprotein catabolic process, and positive regulation of vesicle fusion

Blue—positive regulation of oxidative stress-induced intrinsic apoptotic signaling pathway, methylglyoxal metabolic process, glutamate catabolic process, positive regulation of plasminogen activation, and removal of superoxide radicals.

Red—membrane to membrane docking, neurotransmitter receptor internalization, axon extension, and regulation of actin filament depolymerization

The network of biological processes was made by ShinyGo 0.77, in which these proteins are involved, and this is shown in Figure 6.

As is known, TLE is accompanied by the sprouting of mossy fibers.

Remodeling of neuronal networks is a typical picture for TLE; a number of proteins involved in these processes can be distinguished.

NRP2 is a protein specific to the hippocampal dentate gyrus [60], and it regulates axonal orientation [61], axon pruning, dendritic spine remolding, and other neuroplasticity modulations [62,63] during nervous system development. Nrp2 is involved in the morphogenesis of dendritic spines of excitatory neurons and inhibitory migration of interneurons during development [64]. In addition, polymorphisms at the Nrp2 locus have been found in humans with autism, and Nrp2 knockout (KO) mutant animals exhibit increased susceptibility to seizures [65]. Hyperexpression was observed both in GM tissues and in experimental models, demonstrating the remodeling of dendritic spines and the neurogenesis characteristic of TLE.

DPYSL2, a member of phosphoproteins found in the cytosol, is involved in processes associated with neuronal migration, neuronal polarity development, and axonal growth and direction [66]. DPYSL2 is also associated with comorbid mental illness in Alzheimer’s syndrome [67]. An increase in the expression profile of this protein in the hippocampus of patients with mTLE may be evidence of the processes of synpathic reorganization and plasticity, and it also indicates the processes of neurogenesis, which has already been reported by Curia et al. [68]. Considering its role in the growth and conduction of axons, it is suggested that it may be involved in the process of neuronal sprouting and the generation of seizures [36]. There are some discrepancies in the expression of DPYSL2, as in samples of the hippocampus of patients, it is overexpressed, while the study of animal models indicates its decrease. In addition to being associated with neurodegenerative diseases, DPYSL2 is also a biomarker of immune infiltration in lung adenocarcinoma [69].

GFAP is a marker of astrogliosis [70], characteristic of hanging epilepsy [71]. However, high levels of GFAP are also characteristic of Alzheimer’s disease [72], multiple sclerosis [73], and frontotemporal dementia [74]. Additionally, in general, astrogliosis is not a process strictly specific for TLE [75]. Therefore, its use as a potential TLE biomarker is questioned.

There are data on the association of NME1 with the processes of neuronal development as a result of brain damage, confirmed in in vitro studies and in the study of experimental models [76]. NME1, as a component of the cytosol, can interact with components and regulators of the cytoskeleton: intermediate filaments, actin-binding proteins, adhesive junction proteins, and focal adhesion proteins [77]. In a study by Wright et al. [78], it was found that NME1 is a growth stimulator of dorsal root ganglion neurites, since it may be active as a positive chemotactic signal. Additionally, NME1 can influence the development of the CNS through stimulation of axonal branching [78]. In addition, there is evidence [79] that NME1 has a regenerative and neuroprotective function: its secretion is not observed under normal physiological conditions. Lescuyer et al. [80] found an increase in NME1 expression in the cerebrospinal fluid within six hours after death, which may indicate its role as a marker of neurodegeneration.

A number of proteins are involved in the regulation of neurotransmitter release.

SYT1 is an important regulator of the fast, synchronous, and calcium-dependent release of neurotransmitters; it also modulates synaptic vesicle endocytosis [81,82]. SYT1 has been investigated as a candidate biomarker of synaptic dysfunction and neurodegeneration for a number of neurological disorders [83]. SYT1 is an important modulator of calcium-induced neurotransmitter release [84]. Violations in the work of synaptic transmission can lead to various dysfunctions in the central nervous system. Among these are autism spectrum disorders [85], which are often associated with the presence of mutations in the genes of postsynaptic adhesion molecules [86]. SYT1 is a biomarker for Alzheimer’s disease [87].

There is a large amount of evidence that the activation of the immune response plays an important role in the pathogenesis of TLE [88]. Complement cascade molecules are important components of immune regulation [89,90]. The complement system consists of approximately 35 proteins, including soluble, as well as membrane-bound proteins, whose activation results in a complex cascade of processes leading to microglial activation, secretion of pro-inflammatory cytokines, macrophage recruitment, phagocytosis activation, and increased vascular permeability. It is increasingly recognized that complement activation in the CNS is associated with exacerbation and progression of tissue damage in various degenerative and inflammatory diseases [91,92], as well as in some cases of cancer [93]. However, as is known, neuroinflammation is not specific, and it is observed in a variety of neurological disorders [94], so the use of neuroinflammation proteins as markers is controversial.

However, we found invariance in C4B dysregulation in several TLE profiling studies. However, aberrant expression of this protein is characteristic not only for TLE, but also for other forms of epilepsy: Lafort’s disease [95], as well as febrile seizures [96].

Oxidative stress (OS) is a condition that occurs when the steady state balance of prooxidants and antioxidants shifts towards the former, creating the potential for organic damage [97]. Because H_2_O_2_ is a source of reactive oxygen species (ROS), such as superoxide (O_2_^−^) and hydroxyl radical (OH^−^), which lead to oxidative stress, an endogenous H_2_O_2_ scavenging system is essential for cell viability. PRDX6, an antioxidant protein, is a bifunctional enzyme with the activity of glutathione peroxidase and phospholipase A2 [98]. It is believed that the restoration of the integrity of the membrane is its main function, carried out both by the restoration and by the breakdown of oxidized lipids. It is characteristic for AD [99], childhood cortical dysplasia [100], and Rolandic epilepsy [57].

APOE is involved in AD [101]. APOE isoforms affect the structure and function of mitochondria, which probably leads to changes in oxidative stress, synapses, and cognitive function [102].

GAPDH is a component of the glycolytic pathway [103]. In addition, it is also a regulator of apoptosis [104]. Intranuclear accumulation is characteristic of Parkinson’s disease [105]. GAPDH expression is increased in epileptic tissues [106]. In addition, modifications of GAPDH, such as phosphorylation and S-glutathionylation, alter the GABA signaling pathway under conditions of attack-induced oxidation by regulating gene expression at the post-transcriptional level [106]. It is involved in a number of diseases, such as AD [107], as well as Huntington’s disease [108].

Additionally, a number of other proteins have also been shown to be involved in a variety of neurodegenerative diseases. For example, PRDX2 is characteristic of Parkinson’s disease [109] and depression [110]. BAX is for diabetes-induced non-degeneration [111] and Parkinson’s disease [112]. SLC25A4 is for retinal neurodegeneration [113].

We have analyzed the diseases associated with the studied proteins and their genes. The results are shown in Figure 7.

We also conducted a global search in the DisGenet database, and most of the studied proteins are also involved in the development of AD, PD, MS, etc. A complete summary table of associations is presented in Appendix A.

Only one of the studies we reviewed [56] made an attempt to compare the proteomic profiles of patients with TLE with the proteomic profiles of another disorder, and not just healthy controls. The search for specific biomarkers, in our opinion, must inevitably turn to the discrimination of TLE from other neurological disorders due to the commonality of many histological, cellular, and molecular changes between them. The biomarker of the target disease should help in the discrimination of similar neurological symptoms, and there are classical methods, such as EEG, MRI, clinical tests, etc., to distinguish patients from healthy people.

Additionally, the search for changes in protein expression that are invariant for the stage seem important to us. Altered expression must be significantly maintained over different time periods in order to have good discriminating power. However, in the case of TLE, there is some difficulty in distinguishing the stages of the development of the disease. Thus, the development of TLE is usually divided into three stages: acute, latent (aka epileptogenesis), and chronic (when the disease has already formed) [114]. The acute phase of epilepsy is characterized by the activation of acute phase reactants [115], which are not specific for epilepsy [116,117], but characterize the inflammatory process as a whole. The distinction between the latent and chronic phases is conditional because epileptogenesis includes both the latent period between the provoking event and the onset of seizures, as well as the progression of already formed epilepsy [6]. Therefore, it seems most important to us to study protein expression both in the latent and chronic phases and to search for their invariants.

To date, many studies have focused on the study of the neuroinflammatory response in TLE [118]. However, neuroinflammation is not specific to epileptic disorders only [119]. This is also characteristic of oxidative stress [120].

Compensatory proteins of the response to processes, inflammation, stress, neurodegeneration, etc. also cannot be fully considered specific.

Ultimately, an epileptic seizure is provoked by blocking inhibitory and/or activation of excitatory conduction [121], so it seems important to us to search for protein associations with these mechanisms.

DPYSL, which is involved in the reorganization of the neural network [66], as well as SYT1, which regulates the release of neurotransmitters [81,82], meet these requirements. STMN1, a microtubule regulator, is involved in the process of axonal reorganization during neurogenesis [122], as well as NME1, which is also a suitable candidate. Meanwhile, GFAP, which is a glial marker of gliosis, is nonspecific and characteristic of a number of neurological disorders [123]. C4B is a component of the complement system, a nonspecific immune response [124], which, in our opinion, should be an exclusion factor.

Additionally, it is important to take into account the age of the patient and the length of the disease because altered expression may not reflect a pathogenic process, but age-related changes. In addition, an increase in the duration of the disease, as a rule, can also affect the expression profile.

Consideration should be given to the therapy being taken by the patients being analyzed. For example, levitiracetam can affect the expression of SOD2 [125].

## 4. Conclusions

In this paper, we pooled data from various protein profiling studies in temporal lobe epilepsy. The analysis of these data showed that the results of various profiling studies differ significantly due to the heterogeneity of the input parameters. The profiles differ depending on the type of tissue, stage, region of the brain or body fluid, and similar factors. Despite the many differences between the profiles we identified, the most common differentially expressed proteins in the studies were: DPYSL, SYT1, STMN1, APOE, and NME1. The most common biological processes for them were the positive regulation of neurofibrillary tangle assembly, the regulation of amyloid fibril formation, the lipoprotein catabolic process, the positive regulation of vesicle fusion, the positive regulation of oxidative stress-induced intrinsic apoptotic signaling pathway, the removal of superoxide radicals, membrane-to-membrane docking, neurotransmitter receptor internalization, axon extension, and the regulation of actin filament depolymerization

The main differences and problems in proteomic profiling studies are the large fragmentation of samples for analysis, which can lead to ambiguous and uninformative results. It is necessary to take into account the type of epilepsy (with or without histological changes, drug-sensitive or drug-resistant), as well as the clinical characteristics of patients, since the proteomic profile may reflect not epileptic activity, but processes are not associated with the presence of the disease.

Further study of the proteome of patients with TLE is an important and urgent task. The search for biomarkers of the disease will facilitate diagnosis and allow the detection of disorders in the functioning of neurons at the molecular level, which would make it possible to outline a targeted therapy strategy.

However, MS-based proteomic profiling for relevant research must accept a number of limitations, the most important of which is the need to compare different types of neurological and, in particular, epileptic disorders. Such a criterion could increase the specificity of the search work and, in the future, lead to the discovery of biomarkers for a particular disease.

## Figures and Tables

**Figure 1 ijms-24-11130-f001:**
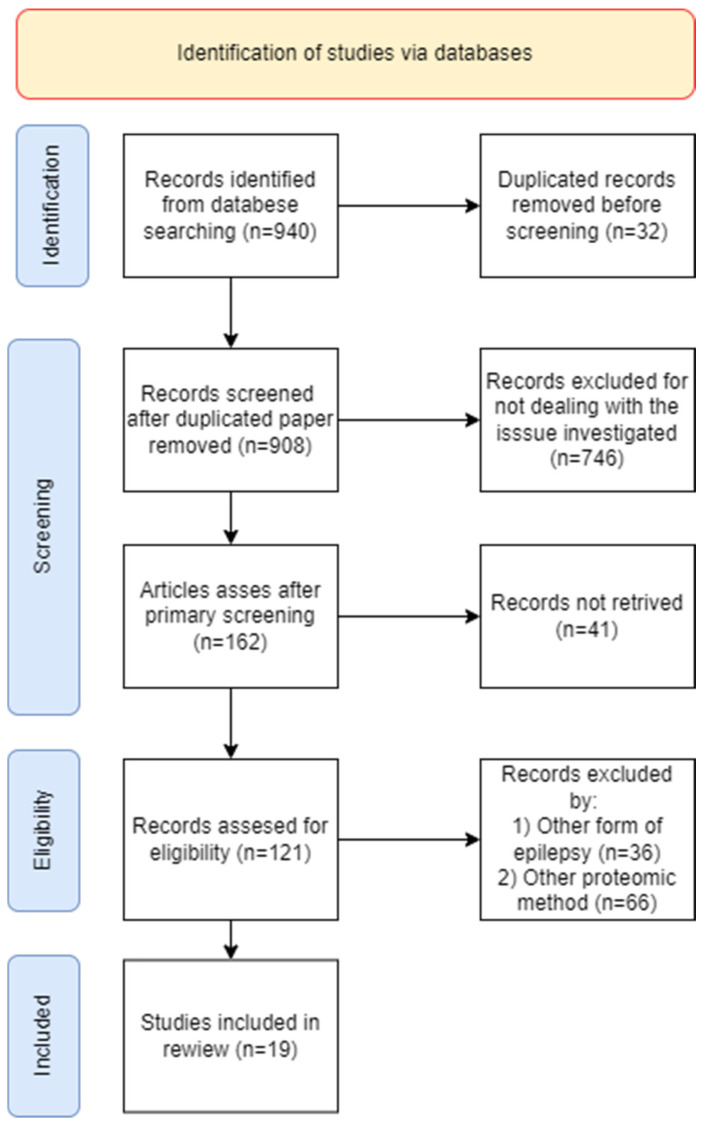
PRISMA chart. The PRISMA flow diagram follows the PRISMA checklist (http://prisma-statement.org/PRISMAStatement/Checklist, accessed on 1 May 2023).

**Figure 2 ijms-24-11130-f002:**
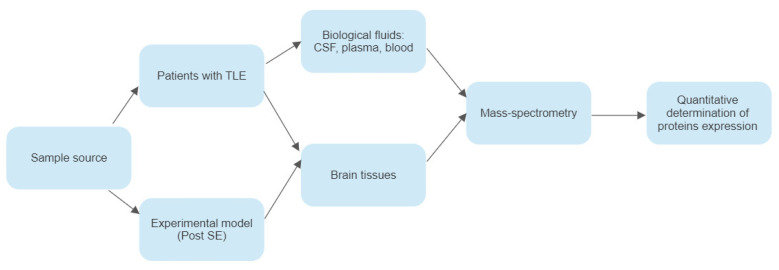
The laboratory methods and samples carried out across the analyzed studies.

**Figure 3 ijms-24-11130-f003:**
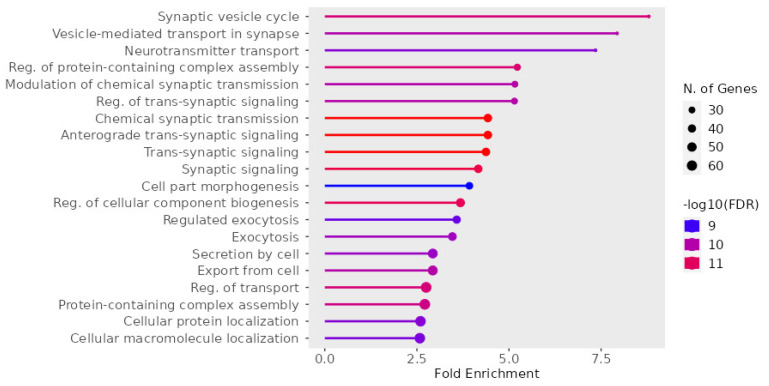
The biological processes of the analyzed tissue proteins [7,22,23,24,32,35,36,37].

**Figure 4 ijms-24-11130-f004:**
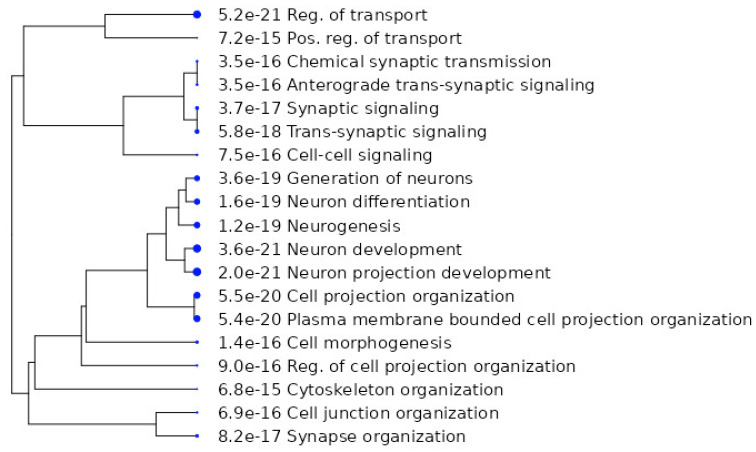
The biological functions of the analyzed [19,20,44,47,48,49,50] proteins.

**Figure 5 ijms-24-11130-f005:**
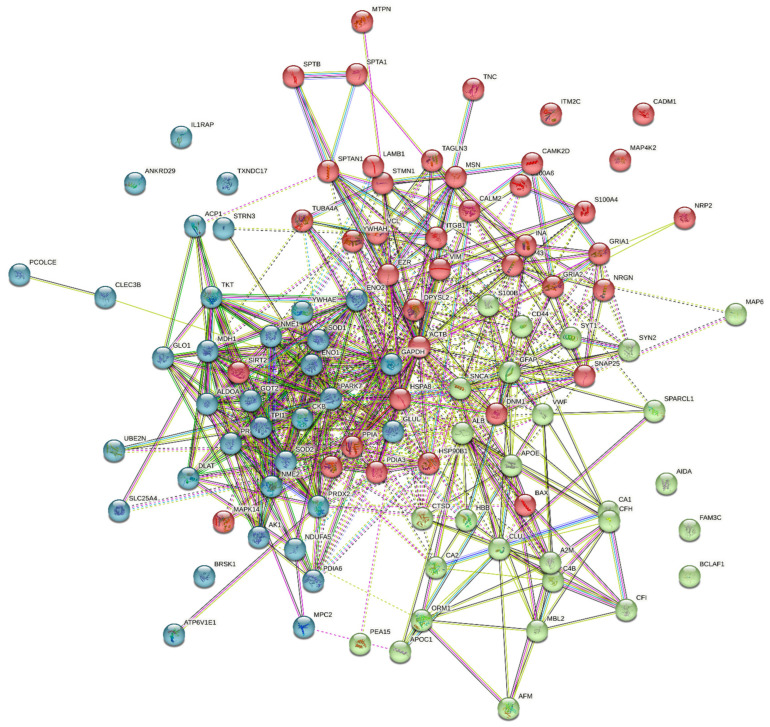
Network of all studied [19,20,22,23,32,35,36,37,44,45,47,50,54,55] proteins.

**Figure 6 ijms-24-11130-f006:**
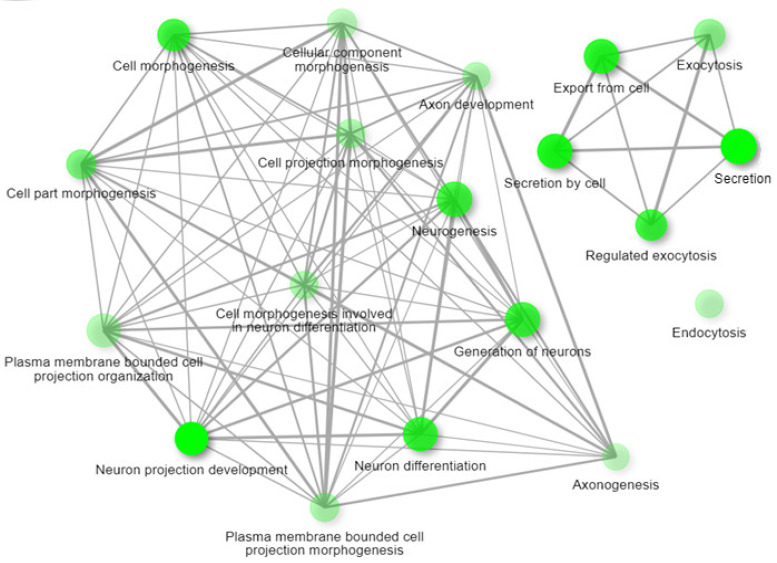
The BP network of the analyzed [19,20,22,23,32,35,36,37,44,45,47,50,54,55] proteins.

**Figure 7 ijms-24-11130-f007:**
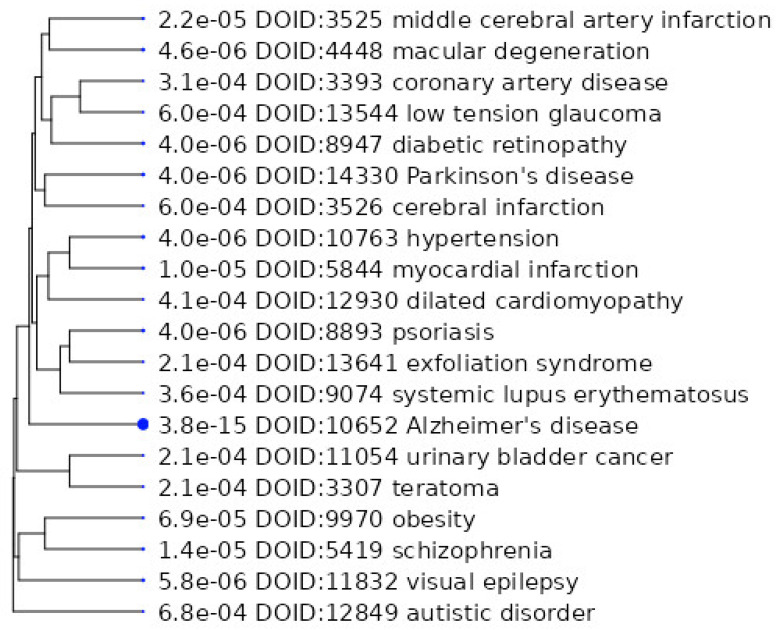
Diseases associated with the studied [19,20,22,23,32,35,36,37,44,45,47,50,54,55] proteins.

**Table 1 ijms-24-11130-t001:** Protein overlap in the different studies.

Publication							
Mériaux [7]	+						
Persike [36]		+					
Liu [22]			+				
Zhang [23]				-			
Yang [35]				-	+	-	
Keren-Aviram [37]		+	-		+	-	
Eun [32]						-	+
Xiao [24]	+						+
Protein	AIDA	DPYSL2	GFAP	SYT1	SNCA	GAPDH	BAX

“+”—increased expression. “-”—reduced expression.

**Table 2 ijms-24-11130-t002:** Overlaps of proteins found in the brain tissues of experimental animals.

Protein	Keck [20]	Bitsika [19]	Qian [50]	Walker [47]	Sadeghi [49]	Araujo [45]	Liu [44]	Xu [48]
Got2	+					-		
Ywhae	-					+		
Park7	-				-			
Pea15	-				+			
Clu	+	+						
Gfap		+	+		+			
Vim		+	+	+	+		+	
Tnc		+					+	
Msn		+					+	
Syn2		-				-	+	
Krt8			+				+	
Nrgn			+				+	
Dnm2				+			+	
Alb				+			+	
Tagln3					+		+	
Glo1					+		+	
Snap25					-		+	
Stmn1					-		+	
Eno2					+		+	
Dnm1					+	+		
Nme1					+		+	
Hspd1					+		+	
Ina					-		+	
Glul					+	-		
Ckb					+	-		
Pdia3					-	+		
Tpi1					+	-		
Syn2		-				-	+	
S100a4							+	+

“+”—increased expression. “-”—reduced expression.

**Table 3 ijms-24-11130-t003:** Protein overlaps.

Overlaps of Circulating Proteins								
Banote [55]	+	+	-	-				
Sun [57]	-	-		+	-	+	+	+
Xiao [54]			+		-	-		+
Hamrah [56]							+	
Protein	MBL2	VWF	APOE	APOC1	SOD1	CTSD	ORM1	PCOLCE

“+”—increased expression. “-”—reduced expression.

**Table 4 ijms-24-11130-t004:** Overlaps of all proteins obtained in the publications.

Protein	Persike [36]	Liu [22]	Zhang [23]	Yang [35]	Keren-Aviram [37]	Eun [32]	Banote [55]	Sun [57]	Xiao [54]	Keck [20]	Bitsika [19]	Qian [50]	Walker [47]	Sadeghi [49]	Araujo [45]	Liu [44]
DPYSL2	+				+										-	
PARK7	+							-		-				-		
GFAP		+			-						+	+		+		
SYT1			-	-											-	
C4B			+					+			+					
PRDX6				+				-						+		
GAPDH				-	-	-		-						+		
STMN1					+									-		+
APOE							-		+		+					
NME1							-							+		+
TPI1								-						+	-	
YWHAE								-		-					+	
MSN								-			+		+			
CLU									-	+	+					
Vim											+	+	+	+		+
Syn2											-				-	+

“+”—increased expression. “-”—reduced expression.

## Data Availability

Not applicable.

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
