# Peer review of "Mass Spectrometry as a Quantitative Proteomic Analysis Tool for the Search for Temporal Lobe Epilepsy Biomarkers: A Systematic Review"

_ijms, 2023, doi:10.3390/ijms241311130_

Round 1
Reviewer 1 Report
May I say that the figures you provide seemed very helpful in summarising what the various studies utilising mass spectrometry had shown.
There are a few matters that I would like to raise with you.
1. You repeatedly indicated the type of mass spectrometer used in individual studies. Was there any special reason for doing this such as particular types of instrument being better able to serve particular purposes, or was the information provided simply for completeness? A reader might wonder.
2. Was the denominator used in expressing the quantities of the various molecules measured the same in all studies? Was it simply wet or dry weight of tissue, volume of tissue, total protein or DNA content, number of neuronal nuclei in a particular tissue volume, or something else.
3. It seems that much of the material you described was derived from chronic, treatment-refractory, mesial temporal lobe epilepsy hippocampus. Can you exclude the possibility that the protein alterations found simply reflect the histological alterations of hippocampal sclerosis rather than having any primary role in seizure genesis.
4. If abnormal proteins, or abnormal protein concentrations were found in circulating blood, can you exclude the possibility that these materials were derived from peripheral tissue damage during seizures, particularly convulsive ones?
The English is a little awkward in a few places. but not enough to be troublesome. The number of acronyms can be overwhelming to a eader not working in the area of the paper.
Author Response
Dear reviewer! Thank you very much for your comments and recommendations!
- The type of ionization in mass spectrometry can significantly affect the safety of the studied sample. Since proteins are quite labile, it seemed important to us to indicate the type of ionization so that the reader can have an idea about the validity of the data obtained in the studies. We have added the sample ionization type to paragraphs where it was not specified. All changes made are highlighted in yellow.
The studies used such types of ionization as: ESI (electrospray), MALDI (matrix-assisted laser desorption/ionization), API (Atmospheric Pressure Ionization), which are soft types of ionization.
- The following denominators were used to express differences in expression: fold change, and ratio of molecular weight to isoelectric point (pI/Mw).
- This is a really important note, thanks. Indeed, proteins found in the brain tissues of patients suffering from hippocampal sclerosis and drug resistance may reflect only histological changes, and not affect or reflect seizure activity. To resolve this conflict, it is necessary to conduct further studies of brain tissues without histological changes, which can be realized by studying model organisms, such as. Similar restrictions were indicated by us in lines 198-207. We have also added another paragraph about these limitations at the end of the tissue profiling paragraph.
- Epileptogenesis and seizure activity can indeed lead to damage and dysfunction of the blood-brain barrier, so we can indeed detect proteins specific to vascular endothelial tissues. However, BBB dysfunction is a typical event in epilepsy, therefore, on this basis, we cannot exclude the study of these proteins as potential biomarkers of epilepsy and/or epileptogenesis.
- With regard to damage to the peripheral nervous system. Indeed, proteins-markers of tissue leakage of the peripheral system can be detected in plasma. Nevertheless, in our opinion, such lesions can be considered an indicator of seizure activity, which does not exclude the possibility of their study as biomarkers.
Best regards, authors
Reviewer 2 Report
The authors have done a very important job in compiling the various publications aimed at describing the expression profiles of proteins associated with temporal lobe epilepsy. Although this article is relevant and deserves to be published, it seems that conclusions are difficult to draw due to the diversity of publications in many respects. The main differences between studies (or inaccuracies) are, for tissue analysis (type of tissue, type of controls, age, sex, patient treatment often not mentioned, etc.), experimental models (type of models, type of studies, time of sampling), analysis of peripheral samples (age, sex, patient treatment often not mentioned, type of controls, differences between plasma and CSF, presence of other diseases...). In addition, studies based on tissue samples are often carried out using tissue from patients with drug-resistant epilepsy, which is not the case for the other two types of study.
These limitations should be one of the main conclusions of this systematic review and should motivate experts in the field to produce guidelines aimed at defining the type of studies to be carried out in the future to improve knowledge.
Minor:
- Why is reference 56 included? This study was carried out on children without TLE
- Why does Table 4 only present the results of 14 studies, when 19 were included?
- Why, in the conclusions, do the authors cite the following proteins as the most commonly differentially expressed: DPYSL, SYT1, STMN1, APOE, NME1. This does not seem to be reflected in Table 4.
- Lines 532-533: "However, plasma can be problematic due to the wide range of protein concentrations". Please specify.
Author Response
Dear reviewer! Thank you very much for your comments and recommendations!
We have added an additional paragraph to the conclusion based on your recommendations. Changes made are highlighted in yellow.
- Indeed, this study was conducted on children, however, diagnosed with temporal lobe epilepsy. Despite this, we felt it was important to include it in the review as it is the only study that attempted to discriminate epilepsy from another disorder, not just healthy controls.
- Table 4 reflects the intersections of the detected proteins in different studies. We analyzed 19 studies, however, the proteins found in five of them were not duplicated in other studies. The lack of overlaps served as the basis for the exclusion of these five studies from the summary table.
- The proteins presented in Table 4, when further analyzed for associations with other neurological diseases, did not show their specificity in inducing and reflecting epileptogenic activity, which is described in the Discussion section. Therefore, we cannot offer them as biomarkers.
- Blood plasma contains more than 3000 different proteins, therefore mass spectral analysis can identify a large number of protein molecules that may not be associated with the disease in any way, but be plasma resident or reflect physiological changes in the body, which complicates subsequent analysis by the redundancy of data that we we can get.
Best regards, authors
Round 2
Reviewer 2 Report
The paper can now be published.